# Integrative Transcriptomic and Metabolomic Analysis at Organ Scale Reveals Gene Modules Involved in the Responses to Suboptimal Nitrogen Supply in Tomato

Begoña Renau-Morata [1], Rosa-Victoria Molina [2], Eugenio G. Minguet [2], Jaime Cebolla-Cornejo [3], Laura Carrillo [4], Raúl Martí [3], Víctor García-Carpintero [5], Eva Jiménez-Benavente [2], Lu Yang [4], Joaquín Cañizares [5], Javier Canales [6,7], Joaquín Medina [4,*] and Sergio G. Nebauer [2,*]

1 Departamento de Biología Vegetal, Universitat de València, 46022 València, Spain; begonya.renau@uv.es
2 Departamento de Producción Vegetal, Universitat Politècnica de València, 46022 València, Spain; rvmolina@bvg.upv.es (R.-V.M.); egomezm@ibmcp.upv.es (E.G.M.); evajibe25@gmail.com (E.J.-B.)
3 Joint Research Unit UJI-UPV Improvement of Agri-Food Quality, COMAV, Universitat Politècnica de València, 46022 València, Spain; jaicecor@btc.upv.es (J.C.-C.); raumarre@upvnet.upv.es (R.M.)
4 Centro de Biotecnología y Genómica de Plantas, INIA-CSIC-Universidad Politécnica de Madrid, 28223 Madrid, Spain; carrillo.laura@inia.es (L.C.); lu.yang@alumnos.upm.es (L.Y.)
5 Bioinformatic and Genomic Group, COMAV-UPV, Universitat Politècnica de València, 28223 València, Spain; vicgarb4@upvnet.upv.es (V.G.-C.); jcanizares@upv.es (J.C.)
6 Institute of Biochemistry and Microbiology, Faculty of Sciences, Universidad Austral de Chile, Valdivia 5110566, Chile; javier.canales@uach.cl
7 ANID–Millennium Science Initiative Program, Millennium Institute for Integrative Biology (iBio), Santiago 8331150, Chile
* Correspondence: medina.joaquin@inia.es (J.M.); sergonne@bvg.upv.es (S.G.N.)

**Abstract:** The development of high nitrogen use efficiency (NUE) cultivars under low N inputs is required for sustainable agriculture. To this end, in this study, we analyzed the impact of long-term suboptimal N conditions on the metabolome and transcriptome of tomato to identify specific molecular processes and regulators at the organ scale. Physiological and metabolic analysis revealed specific responses to maintain glutamate, asparagine, and sucrose synthesis in leaves for partition to sustain growth, while assimilated C surplus is stored in the roots. The transcriptomic analyses allowed us to identify root and leaf sets of genes whose expression depends on N availability. GO analyses of the identified genes revealed conserved biological functions involved in C and N metabolism and remobilization as well as other specifics such as the mitochondrial alternative respiration and chloroplastic cyclic electron flux. In addition, integrative analyses uncovered N regulated genes in root and leaf clusters, which are positively correlated with changes in the levels of different metabolites such as organic acids, amino acids, and formate. Interestingly, we identified transcription factors with high identity to TGA4, ARF8, HAT22, NF-YA5, and NLP9, which play key roles in N responses in Arabidopsis. Together, this study provides a set of nitrogen-responsive genes in tomato and new putative targets for tomato NUE and fruit quality improvement under limited N supply.

**Keywords:** nitrogen; suboptimal conditions; metabolome; transcriptome; growth; leaves; roots; *Solanum lycopersicum*; tomato

## 1. Introduction

Nitrogen (N) plays a crucial role in crop yield and quality [1,2]. Quantitatively, N is the most important mineral nutrient taken up by the plant and is a limiting factor in plant growth and development [3]. Since the 1950s, the use of N fertilizers has been increasing steadily to boost agricultural production and meet global food demands [4]. The use of nitrogen fertilizers worldwide exceeded 117 million tons in 2019 [5] and is projected to

increase to 236 Mt by 2050 [6]. These inputs have led to environmental pollution, climate change, and, indirectly, biodiversity loss [7]. However, it has been estimated that less than half the N applied is taken up by crops [8], so there is margin for a drop in nitrogen inputs without compromising crop yields. The development of varieties with improved nitrogen use efficiency (NUE) would help to reduce fertilizer applications, lower energy costs, and greenhouse gas emissions and mitigate the consequences of nitrogen loss into soil and water sources [9].

The tomato (*Solanum lycopersicum* L.) is one of the most globally important horticultural crops. In 2019, more than 180 million tons were produced for fresh consumption and processing [5]. Furthermore, the tomato fruit is a good source of lycopene, β-carotene, folate, potassium, ascorbic acid (vitamin C), tocopherols (vitamin E), flavonoids, phenolic compounds, and xanthophylls [10]. During the last few decades, the protected cultivation of tomato crops has become the most efficient system to obtain high quality fresh tomatoes for both domestic and export markets [11]. Nevertheless, large amounts of N fertilizer (up to 250–300 Kg/ha) are used to obtain the highest yields under intensive cropping systems [12]. In order to cope with environmental requirements under sustainability conditions, a reduction of fertilizer use is required. However, a decrease in marketable yield is usually reported when N inputs are reduced [13,14]. Although some studies indicate that fine-tuning the fertigation management in tomato hydroponic culture could limit N inputs while maintaining yield [15,16], the development and use of high NUE genotypes is therefore of great interest [17,18]. Nevertheless, N fertilization also impacts fruit quality since it affects the content of sugars and organic acids [16,19]. However, little is known about the effects on nitrogen compounds such as glutamate or GABA, also related to fruit quality.

NUE is a complex trait that encompasses several physiological processes like N acquisition, assimilation, storage, remobilization, and partition among organs [20,21]. Nitrate and urea are the dominant N supplies in most agricultural soils. Nitrate is taken up by root cells via low- and high-affinity transporters, mainly by the NPF/NRT1 and NRT2 families, respectively [22]. Aquaporins and the active transporter DUR3 have been described to mediate urea uptake in plants [23]. Moreover, ammonium also serves as a nitrogen source and its uptake is mainly controlled by AMT transporters. The reduction of nitrate to ammonium is catalyzed by nitrate and nitrite reductases in the cytosol and plastids, respectively. Glutamine synthetase (GS), glutamate synthase (GOGAT), and asparagine synthetase (ASN) are responsible for the assimilation of ammonium into amino acids [22]. Assimilation can take place in the shoots or roots depending on the species and environmental conditions [24]; thus, amino acids or/and nitrate are transported in the xylem to the shoot. Several nitrogen transporters involved in the loading and unloading of xylem or phloem control nitrogen translocation and partition in the plant. In addition, organic N remobilization and storage are important processes contributing to nitrogen economy [22]. Several specific steps in uptake and assimilation have been proposed as potential targets for biotechnological NUE improvement in crops [25]. In addition, efforts are being directed toward the identification of the molecular basis of the transcriptional regulation of NUE. Genomic and global transcriptomic analyses has allowed for the identification of key regulatory genes like *DOF1*, *NLP7*, and *NAC2* to improve NUE in different crop plants [26].

In the case of tomato, little work has been done to address the molecular and physiological traits associated with differences in NUE. However, different cultivars with contrasting NUE have been identified and characterized [18,27]. Notably, the differences in NUE have been related to morphology characteristics such as root length and thickness, and different expression of nitrogen *SlNRT2.1/NAR2.1* and *SlNRT2.3* transporters. Furthermore, tomato $NO_3^-$ transporter genes, belonging to the *NRT1*, *NRT2*, and *AMT* families (*LeNRT1.1*, *LeNRT1.2*, *LeNRT2.1*, *LeNRT2.2*, and *LeAMT1*) and urea (*SlDUR3*) have been identified, but only a few of them have been characterized [18,27–31].

Nitrogen deficiency or starvation triggers important changes in plant growth and development, stimulating root growth relative to shoot growth, and leads to modifications in root architecture [32]. In addition, N deprivation induces changes in the activity of nitrate and ammonium transporters and stimulates the remobilization of N from source organs [33]. Furthermore, the adaptation to the reduced nitrogen supply revealed global metabolomics changes, which showed organ specificity and were time-dependent [34,35]. All these adaptive changes are regulated at transcriptional, post-transcriptional, and posttranslational levels [36,37]. In response to nitrogen, several key transcription factors (TFs) and transcriptional networks have been identified in different plant species. Using different approaches, TFs like NLP6/7, TCP20, TGA1/4, SPL9, or ANR1 have been reported to regulate different genes involved in nitrogen uptake, assimilation, and metabolism as well as growth responses to nitrogen [36,38,39]. In addition, several miRNAs and lncRNAs have been described as being involved in nitrogen regulation in Arabidopsis and crops [40]. Besides, posttranslational modifications through protein phosphorylation have also emerged to play an important role in the regulation of N sensing, uptake, assimilation, and remobilization [41].

Forward genetics and systems biology approaches aim to decipher the complex regulatory networks underlying the responses to nitrogen in plants [39,42–44]. In addition, omics technologies are also contributing to the understanding of the molecular mechanisms and physiological processes regulating N metabolism and homeostasis [35,45]. Several global transcriptomic analyses have been performed to characterize the gene expression responses to N starvation or limited N supply in model plants such as Arabidopsis, and crops like rice, wheat, and oilseed rape [35,45–51]. In addition, studies have been conducted into *Solanum tuberosum* under starvation conditions [52–54]. Every one of these studies revealed shared genes differentially expressed in response to nitrate starvation, which are enriched in processes related to C and N metabolism, transmembrane transport, and photosynthesis. Interestingly, common responses have been reported in other analyses of tomato plants under starvation conditions of other macronutrients such as sulfur and potassium [45,55].

Given the economic relevance of tomato, the identification of physiological processes and key regulatory elements involved in the adaptation to suboptimal N supply would be of great interest for the development of new varieties with improved NUE. In this study, we performed an integrated physiological, metabolomic, and transcriptomic analysis of tomato plants subjected to suboptimal and sufficient nitrogen supply. Our results suggest that different transporters and enzymes related to nitrogen assimilation and distribution are consistently regulated by N availability and revealed new specific biological processes that have not been extensively studied in the context of responses to N including alternative respiration and the photosynthetic cyclic electron flow. We also identified a group of tomato nitrogen transporter genes that might be involved in the remobilization and distribution of nitrogen compounds under limited N supply, and new regulatory factors such as *SlTGA4*, *SlARF18*, *SlHAT22*, *SlNF-YA5*, and *SlNLP9*, which might have a role in the control of long-term N response. Combined bioinformatics and functional analyses of candidate genes will contribute to an improved understanding of the physiology of N responses in plants.

## 2. Materials and Methods

### 2.1. Plant Material and Treatments

Tomato seeds (*Solanum lycopersicum* L. cv. Moneymaker) were surface sterilized with a 20 g/L sodium hypochlorite solution for 5 min, rinsed three times in distilled water, and placed in Petri dishes containing moistened filter paper. After germination, seedlings were cultivated for 15 days in trays filled with vermiculite and fertilized with $\frac{1}{4}$ strength Hoagland no. 2 solution [56]. Plantlets were transferred to pots containing expanded clay (Arlita™, Madrid, Spain) balls (2–3 mm diameter) in growth chamber experiments. Plants were grown at 25/18 °C and 16/8 h (day/night) conditions. It has been shown that 6 mM nitrate in nutritive solution was sufficient to optimize tomato growth from seedlings to early bloom in hydroponic culture [57]. In order to assess the adaptive responses of

tomato to a reduction in fertilization input compatible with sustainable production, we cultivated the plants at the 4 mM nitrogen level (suboptimal conditions) and 8 mM nitrogen as optimal N supply (control). Plants were watered daily with a fertilization solution based on Hoagland and Arnon [56]. Salts were adjusted to maintain the amounts of the remaining essential minerals unaffected.

### 2.2. Analysis of Growth-Related Parameters

Root and shoot biomass were determined every 3–5 days during the 40 days of differential fertilization. Dry matter was obtained after drying the sampled organs at 60 °C for 48 h. Relative growth rate (RGR) was calculated as indicated by Bruggink [58] based on dry matter. The best adjustments were obtained when the data were analyzed in two separate periods: from day 0 of differential fertilization to 15, and from day 16 to 40 (Figure S1). Twenty to fifty different plants (destructive harvest) were used for each treatment and determination time point. Essential elements were determined in shoot and root samples by ICP-OES (macro and microelements) and a CHN elemental analyzer (C and N) at the CEBAS-CSIC Laboratory of Ionomics (CEBAS, Murcia, Spain). Three biological replicates were used for each date and organ.

The assay was repeated in a second experiment and photosynthetic and biomass determinations performed after 20 days of differential N fertilization (8 and 4 mM N). Root and leaf samples were taken and stored at $-80$ °C for metabolomic and transcriptomic analyses. Three biological replicates were used for each organ. Plant material from four independent plants was pooled together for each replicate.

### 2.3. Photosynthetic Determinations

Gas exchange and chlorophyll fluorescence measurements were taken for the determination of photosynthetic related parameters [59] with a LI-6400 (LI-COR, Lincoln, NE, USA). Net photosynthetic rate ($A_N$), stomatal conductance ($g_s$), and actual effective quantum yield efficiency of PSII (PhiPS2) were obtained in steady state conditions and 400 ppm $CO_2$ and saturating light intensity (1200 $\mu$mol/m$^2$ s). The maximum photochemical efficiency of PSII ($F_v/F_m$) was measured in dark adapted leaves. Ten different determinations were performed in mature leaves (fourth leaf from the apex) from different plants. In the same leaf, total chlorophyll and flavonol contents were estimated using a Dualex device (Force A, Paris, France).

### 2.4. Metabolomic Analyses

Primary metabolism components were determined by HPLC-QTOF-SPE-RMN on the Metabolomics Platform of CEBAS-CSIC (Murcia, Spain). Sugar, amino acid, and organic acid contents were determined in leaves and roots after 20 days under differential N fertilization (8 and 4 mM N). For each nitrogen level, three independent extracts were analyzed.

Organic compounds related to organoleptic quality were determined in fruits. Two representative fruits were harvested per plant at the red ripe stage from the third and fourth truss. Three biological replicates were used for analyses. Plant material from three independent plants was pooled together for each replicate. Sugars (fructose, glucose, and sucrose, only found in traces), organic acids (malic and citric acids), and amino acids (GABA and glutamate) were quantified by capillary electrophoresis as described in Cebolla-Cornejo et al. [60]. Sucrose equivalents, related to sweetness perception, were calculated as described in Galiana-Balaguer et al. [61]. The contents of the main polyphenols including the hydroxycinnamic acids caffeic, ferulic, and p-coumaric acid, chlorogenic acid, the flavonol rutin, and the flavanones naringenin and the chalconoid naringenin chalcone, were determined in ripe fruits. Polyphenols were extracted and analyzed as described in Marti et al. [62] by reversed phase liquid chromatography using a fused-core column with an Agilent 1200 Series system (Agilent Technologies, Waldbronn, Germany).

## 2.5. Transcriptomic Analysis

Transcriptomic analysis was performed in the roots and leaves of plants grown at 8 and 4 mM N conditions for 20 days. For each nitrogen level, three independent extracts were analyzed. Plant material from four independent plants was pooled together for each extract. Total RNA was extracted using an RNeasy Plant Mini Kit (Qiagen). One microgram of DNA-free RNA was used to generate poly-A-enriched sequencing libraries by the Genomics Unit of the Central Service for Experimental Research (Universitat de València). Libraries were sequenced at 75-nt paired-end reads on a NextSeq 550 (Illumina). Sequence reads were quality checked using FastQC (v. 0.11.3). Reads were mapped against tomato genome 3.0 using HISAT2 [63] and transcript and gene abundances were calculated using StringTie [64] and the ITAG3.2 gene model. Differential expression analyses between 8 mM and 4 mM N samples were performed using the DESeq2 R package [65]. Genes with an adjusted $p$-value < 0.05 as identified by DESeq2 and absolute $\log_2 FC > 0.5$ were considered differentially expressed genes (DEGs). Gene Ontology (GO) enrichment analyses were performed using AgriGO [66]. The 10 most over-represented biological functions are shown.

Quantitative RT-qPCR analysis was performed following the procedures described in Renau-Morata et al. [59]. Three biological replications and three technical replicates were included for each nitrogen level. The expression levels of the studied genes (Table S1) were calculated according to Livak and Schmittgen [67]. The UBIQUITIN3 gene from *S. lycopersicum* was used as the reference gene.

## 2.6. Differential Expression and Gene Co-Expression Network Analyses

For network analyses, we used the normalized expression data obtained from DESeq2. Low-expression genes (<0.5 tpm in all samples) and the 25% of genes with the lowest variation were filtered out prior to network construction. In this manner, 17,198 and 16,052 genes were used to perform the Weighted Gene Co-expression Network Analysis (WGCNA) [68] for the samples of roots and leaves, respectively. The GWENA R package was used to perform the WGCNA analysis using the Pearson correlation method and signed hybrid network type [69]. The soft power threshold was calculated as the first power to exceed a scale-free topology fit index of 0.8 for each network. In the case of the root network, the beta value was 20 and 16 for leaves. The gene co-expression modules were detected with the R function "detect_modules" implemented in the GWENA R package with a minimum module size of 30 genes and the parameter merge_threshold = 0.7 was used to merge similar modules [69]. The module–metabolites relationships were obtained using the R function "associate_phenotype" implemented in the GWENA R package with a $p$-value threshold < 0.05 and Pearson correlation method [69].

## 2.7. Yield Determinations

To assess yield-associated parameters, plants were grown as indicated above (Section 2.1) and then transferred to 15 L pots containing coconut fiber and cultivated under greenhouse conditions for six months. Plants were watered and fertilized with Hoagland no. 2 solutions containing 8 or 4 mM nitrogen daily. Ten different plants were used for each fertilization level. Chemical pest and disease controls were in accordance with commercial practices. All plant side shoots were removed as they appeared.

Fruits at the red mature stage were harvested until the fourth truss in the greenhouse experiment. Individual fruit weight and total number of fruits per plant were recorded. Shoot vegetative biomass was also measured at the end of the experiment.

Nitrogen assimilation efficiency (NAE) and its components were determined following the methodology by Weih [70], optimized for tomato [71]: N uptake efficiency ($U_N$; g/g) as the ratio between mean plant N content during the main growth period and N in the seed; yield-specific N efficiency ($E_{N,y}$; g/g) as the ratio between fruit yield and the mean plant-internal N content during the main growth period; and fruit yield N concentration ($C_{N,y}$; g/g).

*2.8. Statistical Analyses*

Data were analyzed by a one-way ANOVA using the Statgraphics software (Statgraphics Centurion XVI, Statpoint Tech, Inc., Warrenton, VA, USA). The mean treatment values were compared ($p < 0.05$) by Fisher's least significant difference (LSD) procedure.

## 3. Results

*3.1. Suboptimal Nitrogen Supply Promotes Different Root and Shoot Growth Responses in Tomato*

To limit both the environmental impact and fertilizer costs, nitrogen inputs are being revised to be lowered. Several reports have addressed the responses under N starvation conditions in different crops, but little is known about the integrative metabolic and transcriptomic adaptations under a limited nitrogen supply compatible with sustainable production. In this study, we assessed the long-term responses of tomato plants grown under suboptimal nitrogen supply in growth chambers. Thus, a phenotypic and physiological analysis was performed and tomato (cv. Moneymaker) plants were grown for 40 days under 8 and 4 mM N supply, which have been shown as N non-limiting (8 mM) and limiting (4 mM) conditions, respectively [56,72].

Total biomass and growth kinetics were significantly influenced by nitrogen availability (Figure 1A). After 40 days of differential N supply, plants grown at 8 mM N reached a biomass 1.7 times higher than those grown at 4 mM N. Notably, different impacts were observed on the shoot and root growth and biomass (Figure 1B,C). Nitrogen deficiency first impacted on shoot growth, showing a significant decrease in the N limitation treatment after three days. In roots, no differences in biomass were found after seven days of the experimental procedure (Figure 1C). Interestingly, promoted root growth was observed under the 4 mM N treatment after 10 days, but by day 14, the root biomass was similar to that of control plants grown under the 8 mM N (Figure S1A). Nevertheless, after 20 days of treatment, shoot and root biomass was related to the N supply ($p < 0.05$), showing increasing relative growth rates with the N level (Figure S1B).

To determine whether growth rates were related to the N concentration in specific organs, the nitrogen content was quantified in shoot and root samples during the period of time under study. The results obtained showed a progressive decrease in N content in both shoots (5% to 3% N; Figure 1D) and in roots (4 to 2% N; Figure 1E) under 8 mM N conditions over time. The limitation in N supply provoked a drop in both shoot and root N contents when compared to the plants grown at 8 mM N. The decrease was significant after three days of N limitation in both roots and leaves (Figure 1D,E). It is noteworthy that the early decrease took place before any significant changes were observed in shoot biomass (Figure 1B). Notably, the N content was maintained in shoots and roots from day 20 at each supply level, suggesting that growth rates were adapted to the available N levels.

Since carbon and nitrogen metabolisms are tightly linked in plants [73,74], in order to define the effect of N supply on C metabolism, we determined total C content in shoots and roots using an elemental C analyzer. Measurements were taken in plants after 20 days of differential N supply. Carbon content in the shoot was slightly reduced under suboptimal N conditions (Figure 1F). Interestingly, the different treatments gave rise to no observed differences in C root concentrations (Figure 1G).

In order to further investigate the observed changes in the C content, we determined photosynthetic parameters in the plants subjected to different N supply (Figure 1H–L). Photosynthetic capacity was impaired by the N deficiency (4 mM N), since net photosynthetic rates and PSII effective quantum efficiency were significantly lower (about 64% of controls grown at 8 mM N) after 20 days of treatment (Figure 1H,I). Accordingly, a drop in chlorophyll content was observed in the leaves of plants grown at 4 mM N (39 and 31 $\mu g/cm^2$, at 8 and 4 mM N, respectively). The increased substomatal $CO_2$ concentration ($C_i$, Figure 1J) together with the lower induced stomatal conductance ($g_s$, Figure 1K) suggested non-stomatal limitations to photosynthesis. In addition, a reduction in the maximal PSII quantum efficiency ($F_v/F_m$) was observed in plants grown at 4 mM N (Figure 1L), indicating the occurrence of damage or malfunction in the chloroplast light reactions ma-

chinery. All together, these results confirmed an inhibition of photosynthesis, and thus in C availability for growth under suboptimal N conditions. Furthermore, the smaller amount of C and N available resulted in a drop in biomass production (Figure 1A).

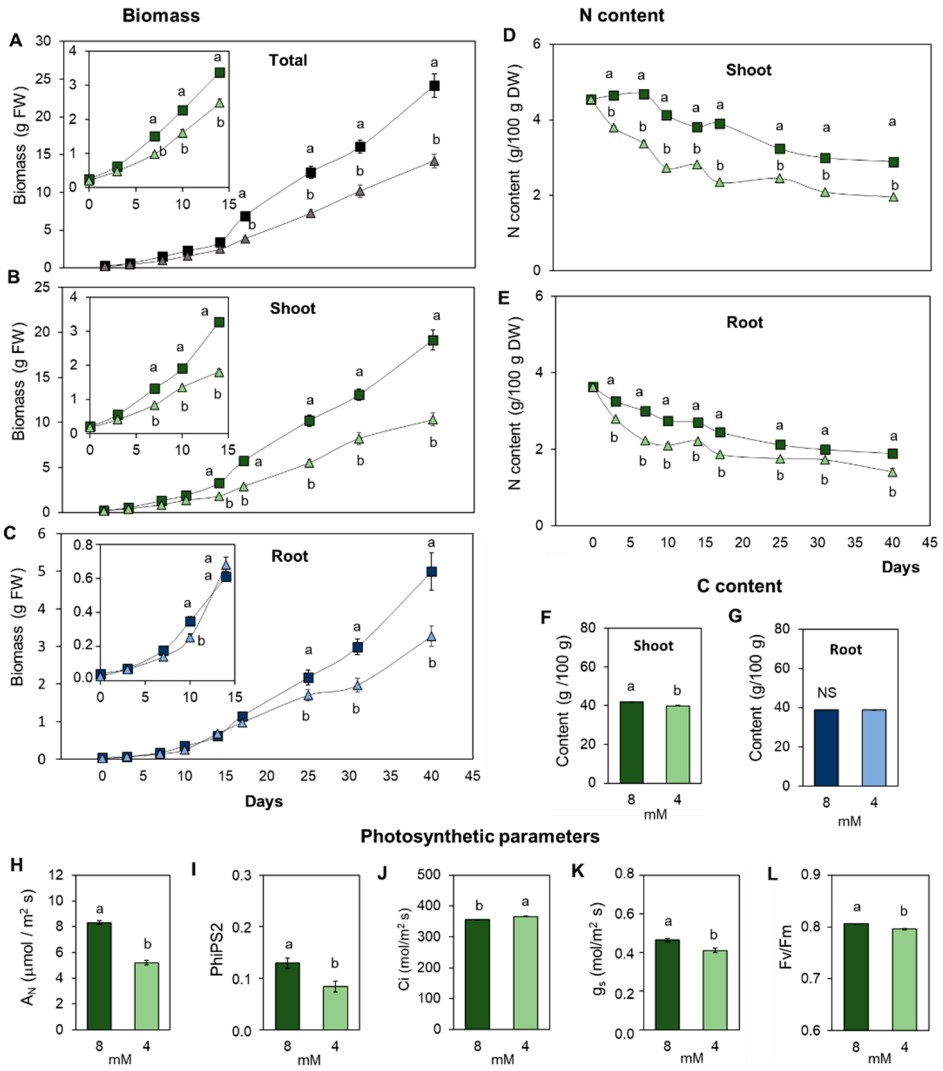

**Figure 1.** Effects of nitrogen supply levels on growth related parameters in tomato. Total (**A**), shoot (**B**), and root (**C**) biomass (g FW) of plants grown under optimal (8 mM N; dark colored bars) and suboptimal (4 mM N; light colored bars) conditions for 40 days. Each value in the mean of 20 to 50 plants. Data during the first 15 days are shown in the small boxes. Total nitrogen in shoots (**D**) and roots (**E**) during the 40 days is shown. Total carbon content in shoots (**F**) and roots (**G**) of plants grown at 8 and 4 mM for 20 days was determined. Values are means ($\pm$SE) of three independent determinations. Photosynthetic measurements were taken: (**H**) net photosynthetic rate ($A_N$), (**I**) effective quantum yield (PhiPS2), (**J**) substomatal $CO_2$ concentration ($C_i$), (**K**) stomatal conductance ($g_s$), and (**L**) maximum quantum yield ($F_v/F_m$). Values are means ($\pm$SE) of ten independent determinations. For each date or determination, different letters indicate significant differences ($p < 0.05$).

### 3.2. The Content of Essential Elements in the Plant Is Differentially Affected in Leaves and Roots by the Limitation in N Supply

Several agronomic and physiological studies reported the existence of close relationships between nutrients [75,76]. To gain insight into the nutritional deficiencies provoked in tomato by the limitation in N supply, we determined the macronutrient (Ca, K, Mg, S, and P) and micronutrient (Fe, B, Cu, Mn, Zn, and Ni) contents in the roots and leaves of plants grown under 8 or 4 mM N supply for 20 days. The results shown in Figure 2 indicated that the reduction in N supply affected the uptake of macroelements. Notably, significant

changes in the contents of several nutrients were observed in an organ dependent manner. In the leaves, while Ca, Mg, and S did not change when using 4 mM N (Figure 2A), P and K contents were reduced under suboptimal N conditions. In contrast, in the macroelements present in the roots (K, P, S, Ca, and Mg) the contents remained unaltered in plants grown under 4 mM N supply (Figure 2B).

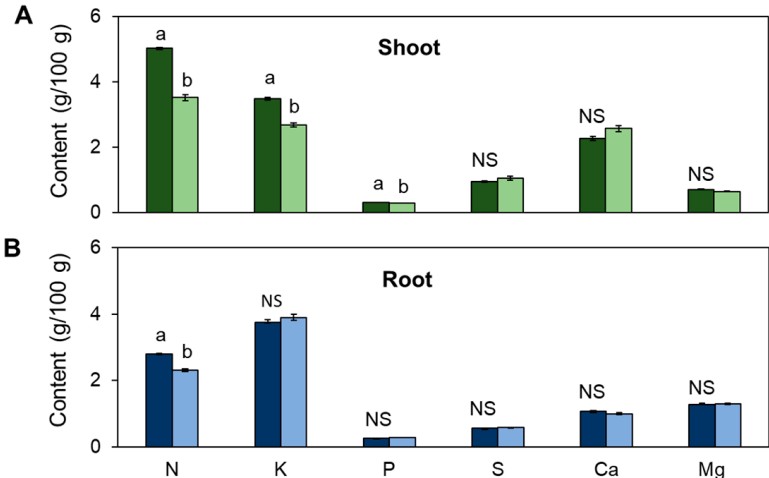

**Figure 2.** Effect of nitrogen supply levels on the uptake of macronutrients. Concentration (g/100 g) of N, K, P, S, Ca, and Mg in leaves (**A**) and roots (**B**) of plants grown under 8 (dark colored bars) and 4 (light colored bars) mM N for 20 days. Values are means (±SE) of 3 independent determinations in different biological replicates. For each nutrient, different letters indicate significant differences ($p < 0.05$).

The limitation in N supply had a differentiated impact on the micronutrient contents in roots and shoots (Figure S2). N availability was not observed to bring about any changes in Ni, Cu, and Mn contents in either the shoots or in roots by the effect of N availability. In leaves, however, although Zn showed a slight increase, the Fe and B contents remained unchanged (Figure S2A). In contrast, the B and Fe concentration increased in roots (Figure S2B). Taken all together, these data suggest that the limitation in N supply impacts mainly on the macroelement contents in shoots. While the K and P concentrations significantly decreased as the N content dropped in the shoots, the macroelement (K, P, S, Ca, and Mg) contents were not affected in roots. Notably, of the analyzed microelements, the Fe and B were significantly accumulated in roots and Zn in shoots under N suboptimal levels.

### 3.3. Suboptimal N Supply Promotes Different Changes in Shoot and Root C/N Metabolism

Water, $CO_2$, and inorganic nitrogen are used by the plant cells to produce sugars, organic acids, and amino acids, the basic building blocks of biomass accumulation [77]. Since plant growth and development depends on nutrient availability, we used HPLC-QTOF-SPE-RMN to quantify the contents of soluble sugars, organic acids, and amino acids in leaves and roots for the purposes of determining the impact of N limitation in the primary metabolism. Moreover, suboptimal N conditions also lead to the accumulation of starch due to the drop in growth and sink activity in the plant [78]. Therefore, to gain insight into the changes in C allocation, we also determined the starch contents in both organs. Analyses were performed in plants grown for 20 days under 8 and 4 mM N levels.

Consistent with the observed reduction in photosynthesis and total C content (Figure 1F,H), the sucrose and glucose contents were reduced in shoots with suboptimal N supply (Figure 3A). In contrast, starch accumulated in leaves when using 4 mM N compared to 8 mM N (Figure 3A), suggesting a drop in sugar export.

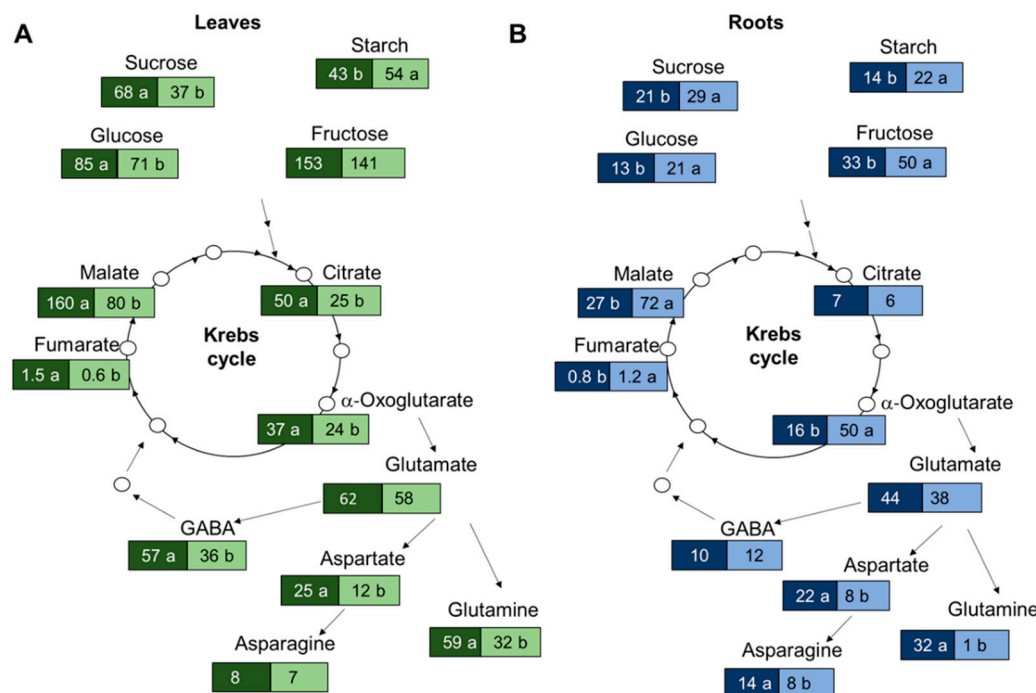

**Figure 3.** Effect of nitrogen supply levels on the primary metabolism. Soluble sugars and starch, organic acids, and amino acid contents (nmol/mg FW) in leaves (**A**) and roots (**B**). Metabolite contents of plants grown under 8 mM N (dark colored bars) and 4 mM N (light colored bars) conditions for 20 days. Values are means (±SE) of 3 independent determinations in different biological replicates. For each metabolite, different letters indicate significant differences ($p < 0.05$).

The results obtained uncovered a reduction in the components of the Krebs cycle such as malate, fumarate, citrate, and α-oxoglutarate in shoots under 4 mM N conditions (Figure 3A). Remarkably, α-oxoglutarate is a key metabolite in N assimilation and precursor for the biosynthesis of key amino acids such as glutamine/glutamate and asparagine/aspartate. Nevertheless, despite the reduction in α-oxoglutarate, no changes in glutamate and asparagine contents were observed under 4 mM N conditions compared to the 8 mM N treatment (Figure 3A). In contrast, the reduction in N supply led to a lower content of aspartate and glutamine. Notably, a significant reduction in the GABA content (Figure 3A), a key metabolite in the nitrogen economy of the plants as well as in that of other amino acids such as alanine, threonine, phenylalanine, tyrosine, isoleucine, leucine, tryptophan, and valine (Figure S3) were also observed under suboptimal N supply.

In the roots, the fructose, glucose, sucrose, and starch contents increased under suboptimal N conditions (Figure 3B). Using the 4 mM N supply, the citrate content did not differ, but malate, fumarate, and α-oxoglutarate increased (Figure 3B). The accumulation of these organic acids might suggest that smaller amounts of precursors were derived from the Krebs cycle for biosynthetic pathways. Accordingly, due to the limitation in N availability, the amounts of glutamine, aspartate, and asparagine were reduced.

Altogether, the results identified specific metabolic rearrangements of organs in responses to long-term N suboptimal conditions. In the leaves of plants grown using 4 mM N, the smaller amount of assimilated carbon was correlated with lower contents of Krebs cycle intermediates and key amino acids such as aspartate and glutamine, although the levels of available asparagine and glutamate levels were maintained. In roots, due to growth inhibition, sugars, starch, and Krebs cycle intermediate levels accumulated under N deficiency. Notably, no significant changes in glutamate and GABA contents were observed.

### 3.4. Nitrogen Limitation Promotes Specific Changes in Leaf and Root Transcriptomes

In order to gain further knowledge about the observed changes in tomato plants under suboptimal N supply conditions (4 mM N), a comparative transcriptomic analysis was

conducted by RNA-Seq. The total RNA was extracted from the leaves and roots of plants subjected to differential nutrition (8 and 4 mM N) for 20 days. We selected this time point for the transcriptomic analyses since plants showed differentiated relative growth rates and N contents at sufficient and suboptimal N fertilization levels (Figure 1A–E).

The leaf transcriptome revealed 1442 DEGs (*p*-value < 0.05; $\log_2$FC > 0.5), of which 981 were downregulated and 461 upregulated (Figure 4A, Table S2). In contrast, only 257 genes were differentially expressed in roots, of which 82 were upregulated whereas 175 were downregulated (Figure 4A, Table S3). Remarkably, a limited number of the differentially expressed genes were shared between roots and leaves. We identified 23 and 38 of the up- and downregulated genes, respectively, which were shared by the roots and leaves (Figure 4A, Table S4). Interestingly, among the shared upregulated genes, several nitrogen compound transporters, starch synthesis genes, and transcriptional regulators were found. Furthermore, well known C/N metabolism genes encoding phosphoenolpyruvate carboxylase 1 (PEPC1), malate synthase, and nitrite reductase, several phosphatases, and different membrane transporters including amino acids and sulfate transporters were downregulated in both roots and leaves.

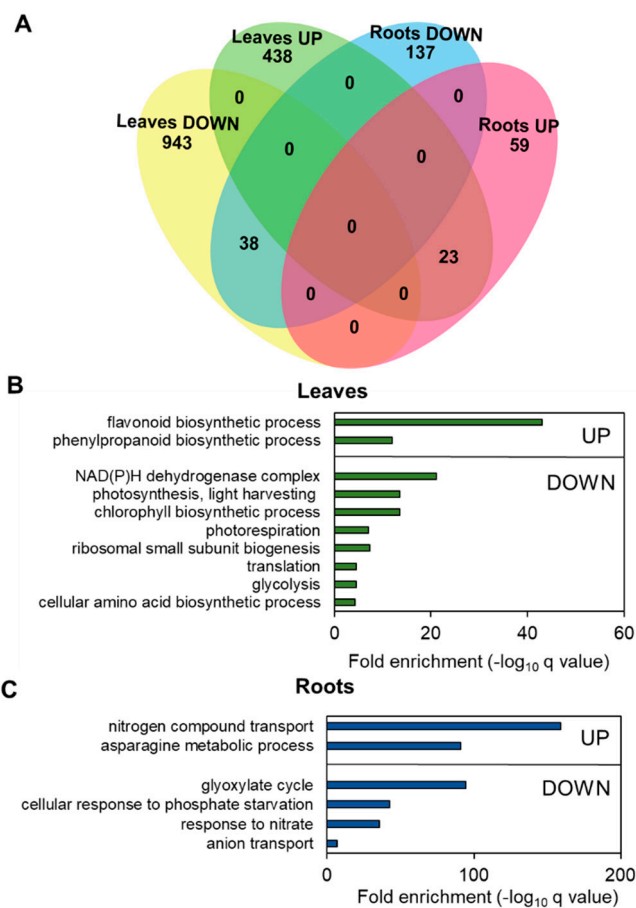

**Figure 4.** Transcriptome analysis of tomato plants in response to nitrogen supply after 20 days under optimal (8 mM N) and suboptimal (4 mM N) conditions. (**A**) Venn diagram showing up- and downregulated genes in leaves and roots. GO term enrichment analysis of up- and downregulated genes by nitrogen in leaves (**B**) and roots (**C**). The most over-represented biological functions are shown.

To verify the transcriptomic analysis, a RT-qPCR analysis of selected DEGs involved in C/N assimilation and partition was performed in leaf and root samples of plants grown under 8 and 4 mM conditions for 20 days (Figure S4). We included sucrose phosphate synthase (*SlSPS*), sucrose transporter (*SlSUT1*), and ADP-glucose pyrophosphorylase

(*SlSADPG*) as key genes involved in sucrose synthesis, sucrose phloem loading and starch synthesis, and nitrate reductase (*SlNR*), glutamine synthetase *(SlGS2)*, and asparagine synthetase *(SlASN2)* as involved in nitrogen assimilation [24,74]. The expression patterns obtained by RT-qPCR were in accordance with the RNA-Seq data, suggesting that similar results could be inferred by both approaches.

An over-representation analysis of biological functions in leaves revealed GO terms that are expected to be conserved in the response to nitrogen limitation and are related to the observed phenotypes such as chlorophyll synthesis, light harvesting in photosynthesis, photorespiration, glycolysis, and amino acid biosynthesis (Figure 4B). Moreover, we found consistent GO terms related to biological processes that have not been as widely studied in the context of the responses to nitrogen availability including ribosome biogenesis, protein translation, and the biosynthesis of phenylpropanoids and flavonoids (Figure 4B). On the other hand, significant enrichment in the categories related to nitrogen compound transport, glyoxylate cycle, asparagine metabolism, responses to phosphate starvation and nitrate, and anion transport (Figure 4C) were found in the root transcriptome.

The most over-represented biological term in roots was 'nitrogen compound transport' (Figure 4C). The re-distribution of nitrogen in the plant is a key process involved in the nitrogen economy of the plant under limiting conditions [79]. We identified several DEGs involved in nitrogen transport in the leaf and root transcriptomes, which included *SlNIT2* (Solyc08g007430.2), *SlNAR2* (Solyc03g112100.3), *SlDUR3* (Solyc08g075570.3), *SlNPF1.17* (Solyc05g005920.2), *SlNPF2.11* (Solyc03g113430.1), *SlNPF2.11* (Solyc03g113430.3), *SlNPF2.6* (Solyc06g075500.3), *SlNPF7.6* (Solyc08g077170.2), and *SlNPF7.3* (Solyc01g006440.1). To further investigate the functions of these genes involved in N transport, we performed RT-qPCR analysis in the leaves and roots of tomato plants grown under 8 and 4 mM N conditions for 20 days. The results shown in Figure 5 indicate that the *SlNIT2*, homologue of the Arabidopsis *NRT1.1* gene, *SlNAR2*, and the urea transporter *SlDUR3* are mainly expressed in roots with sufficient nitrogen supply (8 mM N). *SlNIT2* is downregulated under conditions of nitrogen limitation, whereas *SlNAR2* and *SlDUR3* increased expression levels when using the 4 mM N treatment (Figure 5A–C).

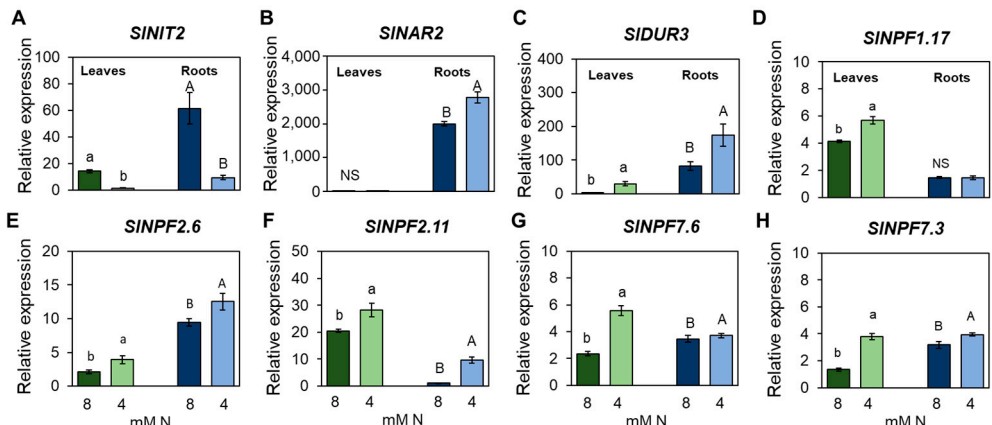

**Figure 5.** Changes in the relative expression of nitrogen compound transporters with different nitrogen supplies. mRNA levels of *SlNIT2* (**A**), *SlNAR2* (**B**), *SlDUR3* (**C**), *SlNPF1.17* (**D**), *SlNPF2.6* (**E**), *SlNPF2.11* (**F**), *SlNPF7.6* (**G**), and *SlNPF7.3* (**H**) genes in leaves and roots under optimal (8 mM N, dark colored bars) and suboptimal (4 mM N, light colored bars) conditions. Each value (±SE) is the mean of three different determinations. For each organ, different letters indicate significant differences (*p* < 0.05).

In addition, different responses were observed in DEGs of the NPF family (Figure 5D–H). *SlNPF1.17* and *SlNPF2.11* showed higher expression in leaves than in roots (Figure 5D,F). In leaves, both genes increased their mRNA levels under 4 mM conditions when compared to 8 mM N. Notably, *SlNPF2.11* is also upregulated in roots under 4 mM N conditions.

Moreover, an increased expression of *SlNPF2.6* is observed in both roots and leaves when using suboptimal N supply (Figure 5E). Furthermore, *SlNPF7.3* and *SlNPF7.6* were specifically upregulated in the leaves under conditions of N supply limitation (Figure 5G,H). Together, all these results might indicate that the identified group of genes encoding nitrogen transporters participate in different responses to nitrogen in tomato, likely displaying specific functions in the studied organs. Interestingly, several transporters are upregulated when N availability is limited, suggesting a role in the redistribution of nitrogen species under conditions of nitrogen limitation in tomato.

### 3.5. Identification of Transcription Factors Involved in the Responses to Nitrogen Limitation at Organ Scale

The transcriptomic analysis revealed major changes in the expression of genes in the leaves and roots of tomato under suboptimal N fertilization conditions (4 mM N). Up to now, complex signaling pathways and regulatory factors have been reported to control nitrogen responses in different plant species [80–83]. In the model plant Arabidopsis, several master transcription factors have been involved in N stress responses such as NLP7, TCP20, TGA4, or TGA1, which regulate the expression of several genes in response to nitrogen [36,38]. However, little is known about the transcriptional regulation and the TFs involved in the responses to N limitation in tomato. The analysis of transcriptomic data allowed for the identification of 67 DEGs in leaves related to TFs of different families (Table S5). MYB and bHLH are the most abundant families with 13 and 8 members, respectively. Interestingly, the homologues of Arabidopsis *TGA4* (Solyc04g054320), *NLP9* (Solyc08g013900), *GATA17* (Solyc12g099370), *HAT22* (Solyc02g063520), NF-YA5 (Solyc08g062210), *NF-YA9* (Solyc01g008490), and *ARF18* (Solyc05g056040) related to nitrogen responses were identified in tomato.

In roots, 18 differentially expressed TFs were found (Table S5), the MYB and G2-like families being the most abundant with three members each. Notably, only *NLP9* (Solyc08g013900), *NF-YA5* (Solyc08g062210.3), and the G2-like *SlHHO6* (Solyc01g108300) genes were shared with the differentially expressed TFs of the leaf, thus suggesting organ-specific regulation in response to N deficit.

These results suggest that both common and specific regulatory networks are activated in tomato roots and shoots under conditions of nitrogen deficiency. Furthermore, the data might indicate that several homologous transcription factors involved in nitrogen responses in Arabidopsis might also display similar functions in tomato.

### 3.6. Transcriptomic and Metabolomic Dynamics in Roots and Leaves under N Limitation Conditions

In order to gain further insights about the patterns of expression of the genes, metabolites, and associated biological processes affected by N limitation treatment in tomato, we performed a weighted gene co-expression network analysis using the GWENA tool [69], which takes advantage of correlations amongst genes and groups genes into modules using network topology [69] and for relating modules to one another and to external sample traits (i.e., phenotypic associations). In this manner we identified 29 and 24 co-expression clusters for genes expressed in roots and leaves, respectively (Figure 6; Tables S6 and S7). In addition, we found that the majority of nitrogen DEGS (85% in roots and 89% in leaves) were contained in clusters 1, 4, and 5 in the roots, and clusters 1 and 2 in leaves (Figure 6; Tables S6 and S7). Notably, these clusters were also correlated with significant changes ($p < 0.05$) in the content of different metabolites analyzed (Figure 6). Thus, we decided to investigate these clusters in more detail.

As shown in Figure S5, the expression profiles of clusters in the roots and leaves showed similar trends. Genes in root clusters 1 and 5, and leaf cluster 1 were repressed under N limitation conditions compared to N sufficient conditions. On the other hand, the expression levels of genes in root cluster 4 and leaf cluster 2 showed significant increases after the N limitation treatment (Figure S5). Although the expression profiles of the genes contained in the identified clusters were similar between roots and leaves (Figure S5), the



identity of these N-responsive genes differed between these organs (Tables S6 and S7). Consequently, we found different enriched biological processes associated with clusters 1, 4, and 5 in the roots and 1 and 2 in the leaves.

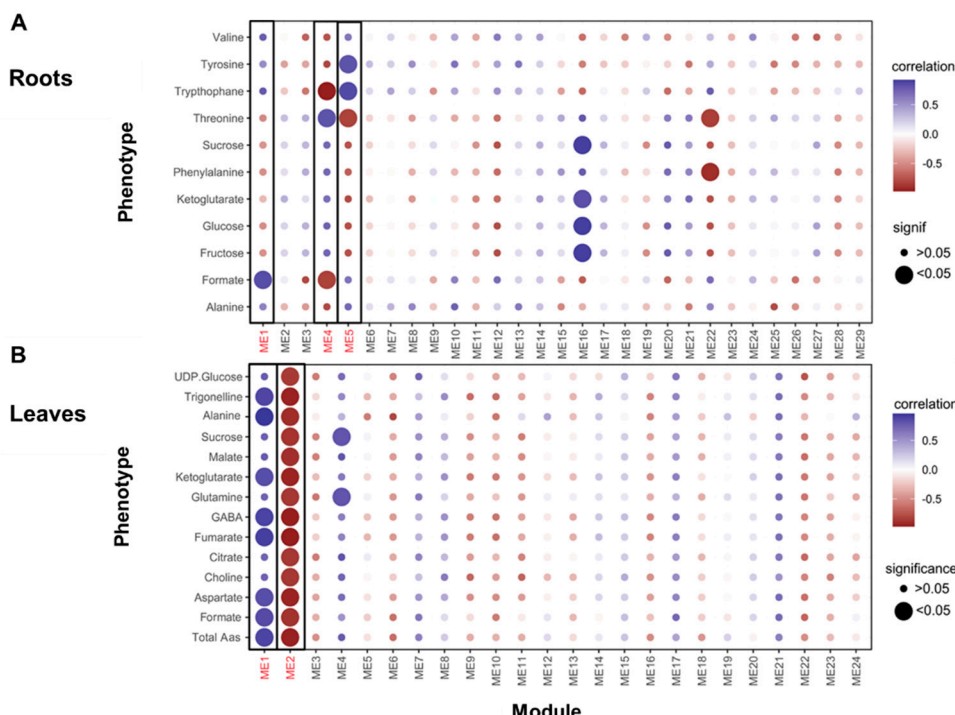

**Figure 6.** Correlation of metabolites showing significant changes in contents to N supply with co-expression clusters of genes expressed in roots (**A**) and leaves (**B**). Co-expression network analysis was performed using the GWENA tool [69] withnormalized expression data obtained from the DESeq2 package. Modules with a significant overlap with the DEGs (*p*-value < 0.05) are highlighted with a black square and red letters.

In the roots, clusters 1 and 5 were enriched in genes associated with C and N metabolism and to oxidative stress and redox activity such as "hydrogen peroxide catabolic process" and "regulation of superoxide dismutase activity" as well as genes related to phosphate responses and transport such as "cellular response to phosphate starvation" and phosphorus metabolic process and ion transport (Table S6).

In leaves, we found that cluster 1 was enriched in genes involved in photosynthesis, TCA cycle, glyoxylate metabolism, and several GO terms related to translation and ribosomal biosynthesis like ribonucleoprotein complex biogenesis or translation ribosome assembly (Table S7). GO terms related to photosynthesis are especially abundant in this cluster, thus we analyzed in more detail the distribution of these genes across different categories of primary metabolism using MapMan annotation framework [84]. In the case of photosynthesis, most genes are involved in light reactions and the Calvin-Benson cycle and sucrose and starch metabolism (Figure S6). In addition, cluster 1 also showed genes related to other aspects of primary metabolism such as N assimilation, amino acid biosynthesis, and TCA cycle (Figure S6). In this regard, as observed in the correlation analyses (Figure 6), the N limitation treatment promotes significant changes in different organic acids (malate, citrate, fumarate, α-ketoglutarate) and different amino acids. In addition, we also identified genes, in the root and leaf clusters, that were also positively correlated with changes in the levels of formate. Notably, formate metabolism is closely related to serine synthesis and other related metabolites, which might suggest that modulating formate levels is part of the response to N limitation.

In leaves, cluster 2 is enriched genes related to RNA metabolism and splicing like RNA splicing and mRNA processing, regulation of mRNA splicing via spliceosome as well as

genes involved in autophagy, proteolysis, and amino acid degradation such as autophagosome assembly, process utilizing the autophagic mechanism, and valine, leucine, and isoleucine degradation and aging. Notably, as can be gleaned from the correlation analyses in Figure 6, extended N limitation altered the levels of numerous metabolites including noticeable changes in compounds linked to amino acid and carbohydrate metabolism, suggestive of a broad increase in recycling.

In summary, N limitation promotes significant changes in the expression patterns of a large number of genes in roots and leaves, many of which are associated with functions related to photosynthesis and nitrogen metabolism and remobilization. In addition, the comparative analysis of major gene co-expression clusters between roots and leaves indicates an organ-specific response of the tomato plants to nitrogen limitation.

### 3.7. Suboptimal N Supply Promotes Significant Changes in Yield and Fruit Quality in Tomato

Improving the efficiency of the use of nitrogen (NUE) for production under sustainable conditions is a major challenge in agronomy and breeding [9]. We reported major transcriptomic and metabolomic changes in response to the limitation in N supply, which led to a significant reduction in plant growth rates. We further wanted to assess the effect of suboptimal N conditions on the use of C/N compounds in the plant during the reproductive growth stage and the impact on the production of fruits. To characterize the processes involved in the use of N for yield, we determined NUE and related parameters [70]. The tomato plants were grown for 140 days under 8 and 4 mM N conditions in greenhouse conditions and ripe fruits were harvested until the fourth truss was developed. Nitrogen assimilation use (NAE), N uptake ($U_N$), and yield N ($E_{N,Y}$) efficiencies and N fruit content ($C_{N,Y}$) were determined [70,71].

The limitation in photoassimilate availability in the source leaves reported when using suboptimal N supply led to a reduction in both vegetative biomass and fruit production (Figure 7A,B). The results shown in Figure 7E indicate that the changes in C and N use under conditions of N limitation provoked a decrease in nitrogen assimilation efficiency (NAE). The lower NAE stemmed from the decrease in all three components: N uptake ($U_N$) efficiency, yield N efficiency ($E_{N,Y}$), and N fruit content ($C_{N,Y}$) components (Figure 7F–H). These results indicate that the limitation of N supply negatively impacts every physiological process involved in the uptake and assimilation of N, the partition of photoassimilates to the fruit, and the fruit N composition.

The analysis of the fruit's C content in plants grown under 4 mM N conditions revealed similar values to those of fruits of plants grown at 8 mM N conditions (Figure 7C). In contrast, the limitation in N supply led to a lower total N content in the fruits (Figure 7D). These data suggest that under suboptimal N conditions, C homeostasis is maintained in tomato fruits. Nevertheless, the drop in N supply caused a reduction in the fruit efficiency for N uptake by the fruit. Since C and N are the main components of the major organic compounds related to fruit quality [59], we characterized the effects of suboptimal N supply on the contents of major organic metabolites determining the organoleptic properties of the fruit. To this end, we quantified organic acids, soluble sugars, and amino acids related to fruit quality in mature red fruits harvested in the plants grown under 8 and 4 mM N under greenhouse conditions (Figure 8). Similar sugar (fructose, glucose, and sucrose equivalents) and citric acid contents were observed in the fruits of plants grown under suboptimal and control N conditions (Figure 8A–D). However, a greater amount of malic acid was observed in the fruits of 4 mM N plants (Figure 8E). In contrast, the main N components of the fruit such as glutamic acid and GABA showed reduced contents in the plants grown using a limited N supply (Figure 8C,G). Altogether, these data indicate that with limited N supply, the tomato plants reduced the total fruit yield (56% of optimal conditions), although the C content in the fruit was maintained when compared to the control plants. Nevertheless, there was a reduction in the N components of the fruit, showing lower concentrations.

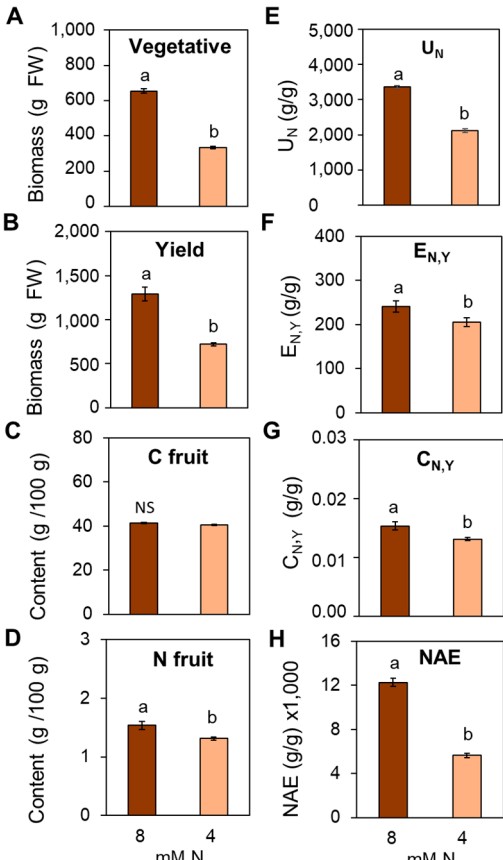

**Figure 7.** Changes in biomass and nitrogen use for yield in tomato plants grown under suboptimal N supply conditions. Vegetative (**A**) and reproductive (**B**) biomass (g FW) in plants grown under 8 (dark colored bars) and 4 (light colored bars) mM N conditions for 140 days in greenhouse conditions. Carbon (**C**) and nitrogen (**D**) concentration (g/100 g DM) in mature red fruits. Values of (**E**) the N uptake efficiency ($U_N$; g/g), (**F**) yield-specific N efficiency ($E_{N,Y}$; g/g), and (**G**) yield biomass ($C_{N,Y}$; g/g) components of (**H**) nitrogen assimilation efficiency (NAE; g/g). Each value ($\pm$SE) is the mean of three different determinations. Different letters indicate significant differences ($p < 0.05$).

The performed GO analyses revealed that phenylpropanoid and flavonoid biosynthesis terms were over-represented in leaves (Figure 4B). Accordingly, a slight increase in total flavonol content was observed in leaves under suboptimal N supply conditions (1.29 vs. 1.14 absorbance units under 4 and 8 mM N, respectively). Since these bioactive compounds are also accumulated in tomato fruits and participate in their functional value [85], we quantified the contents of flavonoids and hydroxycinnamic acids in mature red fruits of the plants grown using 8 and 4 mM N. The analysis showed that the contents of major flavonoid contents such as naringenin and naringenin chalcone were not affected by the N supply (Figure S7A,B). Nevertheless, a significant increase (+50%) in the quercetin, rutinoside, and rutin levels was observed (Figure S7C). In contrast, no significant changes were observed in the concentration of the main hydroxycinnamic acids like caffeic, ferulic, and p-coumaric acids and the derived ester chlorogenic acid (Figure S7F–H). Together, these results confirm minor changes in the phenylpropanoid biosynthetic pathway in tomato fruits under suboptimal N supply conditions, although the changes in the expression of specific genes might lead to a significant accumulation of the bioactive rutin, a major functional compound in tomato.

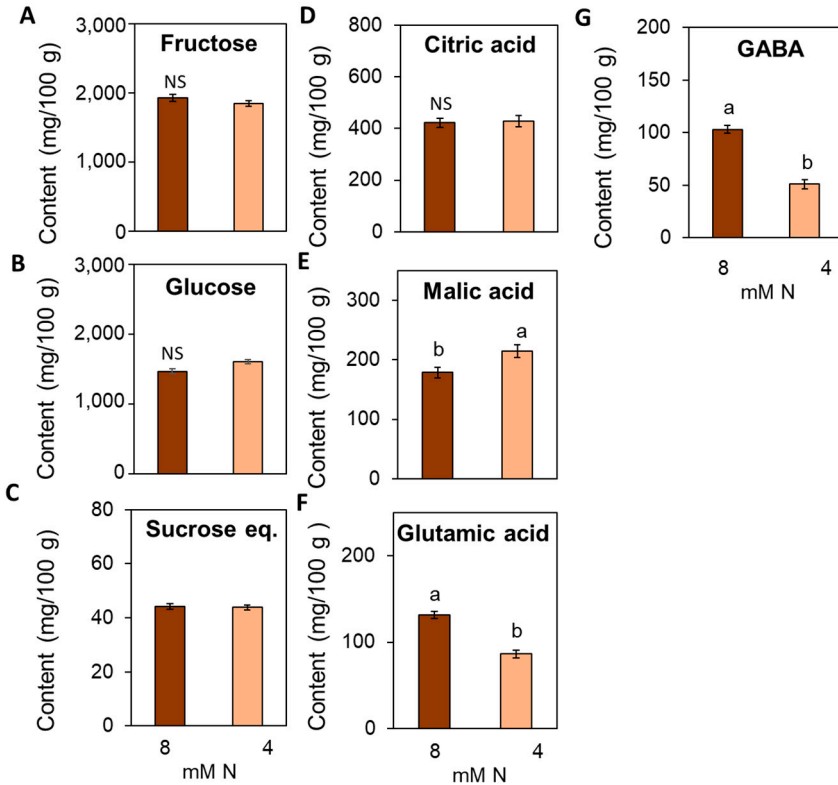

**Figure 8.** Changes in metabolite contents related to organoleptic quality in the fruits of plants grown using 8 and 4 mM N supply. Sugars (**A**) glucose and (**B**) fructose, sucrose equivalents (**C**), organic acids (**D**) citric and (**E**) malic acid, and amino acids (**F**) glutamic acid and (**G**) GABA contents (mg/100 g FW) were determined in mature red fruits. Each value is the mean of three determinations in different fruits. Different letters indicate significant differences ($p < 0.05$).

## 4. Discussion

Both C and N nutrients are essential for many cellular functions and, therefore, an adequate supply of these two nutrients are critical for plant growth, development, and ultimately for the completion of the life cycle and production of harvestable organs. To optimize growth in the ever-changing environment, plants must balance the distribution of C/N compounds between shoots and roots for energy, nutrient, and water resources [86]. This coordination is exerted through root–shoot communications involving multiple signals such as C and N species, hormones, peptides, transcription factors, or RNAs [80,87]. Under conditions of nitrogen limitation, adaptative responses are triggered in the plant, and it has been reported that different strategies are activated specifically in the roots and leaves among species [35]. Up to now, most of the reports have used starvation or conditions of very low N supply to study the responses to nitrogen [35,45,47,49]. However, in this study, we performed detailed transcriptomic and metabolomic analyses of tomato grown under suboptimal N supply conditions (50% of the optimal amount) to identify the processes and genes responsible for the adaptations to N availability. These fertilization levels are closer to the input goals required for sustainable agriculture in the future, and thus, our data will be of great interest for the identification of key players in the development of tomato genotypes with higher NUE adapted to sustainable conditions [88].

### 4.1. Organ-Specific Adaptation of the Primary Metabolism to N Availability in Tomato

The limitation in N supply has severe effects on the C and N primary metabolism in plants, especially on sugars, organic acids, and amino acid contents [35,45,89,90]. It is well-known that under conditions of N limitation, the net carbon gain is reduced as a result of the decrease in total leaf area brought about by growth inhibition, and the drop

in photosynthetic capacity [91,92]. Consistently, our results showed that shoot growth and photosynthetic parameters were downregulated (54 and 63% of optimal conditions, respectively) under suboptimal N supply conditions, and led to a lower total C content (40 vs. 42% of control conditions) in the shoot (Figure 1). Furthermore, it is reported that the decrease in source strength together with the lower sink capacity of other organs markedly alters carbon partitioning as well as allocation. In general, sucrose, hexoses, and starch contents increased in N limited plants [93]. Notably, a similar trend was observed in roots showing higher soluble sugars and starch levels under suboptimal N supply conditions (Figure 3). In leaves, although the starch content slightly rose, the sucrose and glucose contents decreased when using 4 mM N. The observed drop in the sucrose level suggest that under suboptimal N supply, the assimilated C is still used and exported to sustain plant growth. Consistently with this hypothesis, we observed the upregulation of sucrose symporter *SlSUT1*, which is involved in the loading of photoassimilates in the phloem [94] under suboptimal N supply conditions. Moreover, our results also suggest that starch accumulation accounts for the surplus of assimilated carbon, especially in roots that might function as reservoirs for photoassimilates, since a significant increase in root starch and soluble sugars was observed under N limiting conditions.

The assimilated carbon by photosynthesis provides energy and C skeletons for the incorporation of inorganic N into amino acids [95]. Under nitrogen limitation conditions, the total amino acid contents generally decreased, although different adaptative strategies have been reported to be specifically activated in roots and shoots [35,89]. In our study, total amino acid contents also decreased in leaves under conditions of limited N supply (376 vs. 254 µg/mg FW under 8 and 4 mM N, respectively) due to the reduction of most major amino acids (Figure 3 and Figure S3). Interestingly, glutamate and asparagine levels, key amino acids transported in the phloem [96], did not change under suboptimal 4 mM N supply conditions. The total amino acid levels in roots were similar under 8 and 4 mM N supply (84 vs. 82 µg/mg FW). Among the determined amino acids, glutamate levels stayed nearly constant in roots [35,97]. However, although glutamine and aspartate decreased significantly, threonine and phenylalanine accumulated under N deficiency conditions.

We reported reduced available C skeletons such as sugars and organic acids and assimilated amino acids in leaves (Figure 3 and Figure S3), while in roots, C compounds increased but the total amino acids did not change, as reported in other plant species [98,99]. Furthermore, nitrate reductase expression levels (Figure S4D) suggest that most nitrogen is assimilated in the tomato leaves. Therefore, our results indicated organ specific responses in C/N metabolism triggered by N supply limitation. In the leaves, the reduced levels of amino acids under conditions of N deficiency might be derived from the decrease in both the inorganic N and carbon intermediates available for assimilation [89]. However, the amino acid contents in roots should be mainly imported from the shoot as our data suggest no significant N assimilation in this organ. Nevertheless, the high levels of expression of the *SlASN2* gene (Figure S4) in roots under suboptimal N supply conditions might be related to the synthesis of asparagine by incorporating the released ammonium in recycling and remobilization processes [100]. Moreover, asparagine may be transported to the shoot to maintain its homeostasis under conditions of suboptimal N supply. Together, amino acid contents in roots would be determined by the import from the phloem, although the assimilation of released ammonium might also be contributing to the amino acid pool.

Integrative analysis of transcriptomic and metabolomic data revealed significant correlations between different gene clusters and the primary metabolism compounds analyzed in this study in both the leaves and roots (Figure 6). In leaves, we found an enrichment in genes involved in photosynthesis and sucrose and starch metabolism as well as genes related to nitrogen assimilation such as amino acid biosynthesis and TCA cycle. Furthermore, genes involved in autophagy and valine, leucine, and isoleucine degradation were also found in leaves. In addition, the identified clusters of genes in roots were enriched in genes associated with C and N metabolism.

Notably, the expression patterns of several genes in leaf and root clusters were positively correlated with changes in the levels of different organic acids (malate, citrate, fumarate, α-ketoglutarate), different amino acids, and formate levels, which might indicate that these genes are likely involved in the related C/N metabolic pathways. Remarkably, it has been reported that formate is involved in energetic metabolism and responds to stress conditions such as hypoxia, darkness, and heat [101], suggesting that formate metabolic pathways (e.g., photorespiration) are likely involved in tomato responses to N limitation. Altogether, these results suggest that the limitation of N supply promotes changes in the gene expression of many genes associated with functions in C and N metabolism and remobilization that impacts on metabolic rearrangement at the organ scale in tomato.

### 4.2. Long-Term Expression Analyses Identified Several Categories Involved in N Responses in Tomato

Most current understanding of the transcriptomic responses of plants to nitrogen availability is based on results derived from studying the model plant Arabidopsis [35,45,46] and in some crops such as durum wheat [47], wheat [48], rapeseed [51], rice [49], or tea [50] under different conditions of limited N supply. Nevertheless, less information about gene expression responses to nitrogen is available for horticultural crops such as tomato. In this work, we report a comprehensive transcriptome analysis under suboptimal nitrogen supply conditions of this economically important crop. Our survey unraveled major changes in gene expression, especially in leaves. The analysis of the enriched GO terms in leaves revealed biological processes related to photosynthesis, respiration, photorespiration, and amino acid metabolism. Furthermore, glyoxylate cycle, asparagine metabolism, responses to nitrogen and phosphate starvation, and nitrogen transmembrane transport categories were overrepresented in roots. Similar categories have been reported in the above-mentioned transcriptomic analyses of N responses for other plant species [47,48,50,51], thus indicating conserved responses among species. In agreement with these results, the metabolic and physiological responses described in our study confirmed the changes in both the C and N metabolisms and transport under conditions of N deficiency highlighted in the transcriptomic analyses.

We also identified significant enrichment in genes related to the mitochondrial alternative respiration and chloroplastic cyclic electron transport processes among the N responsive categories in tomato. The alternative AOX respiration helps to maintain a redox balance as well as metabolic homeostasis in plants [102]. This pathway has been shown to consume excess sugars and starch in leaves and thus exerts a control over the C/N balance in plants under limited N supply conditions [103]. Moreover, it has been reported that the chloroplastic cyclic electron flow (CEF) is induced under N deprivation conditions in order to readjust the balance between ATP/NADPH for the cellular metabolism [104]. Notably, it is proposed that AOX and CEF interact to optimize protective adaptations under stress conditions [105]. Taken together, our data unraveled the mitochondrial AOX and chloroplast CEF as key processes acting in tomato leaves to optimize redox and C/N balance under N deficiency conditions. Interestingly, these categories have not been previously reported in other crops in response to N [45,47,48,50,51].

Our transcriptomic analysis identified several genes involved in transmembrane transport, and among them, related with the distribution of nitrogen compounds, like nitrate, amino acids, peptides, and urea (Tables S2 and S3). Nitrogen transporters are key players in the plant nitrogen use and partitioning since they regulate root uptake, root-to-shoot transport, leaf-to-sink transport, remobilization, and the storage of inorganic and organic N compounds [22]. Accordingly, many studies have shown that engineered crop plants such as maize or rice with an altered expression of nitrogen transporters displayed improved yield or NUE [106–108]. Several transporter genes involved in nitrate, ammonium, and urea have been identified in different species such as Arabidopsis and rice [7]. In tomato, the role of different genes involved in nitrogen compound transport in the roots has been studied like the low affinity *SlNRT1.1* and high affinity *SlNRT2.1/SlNAR2.1* transporters for nitrate acquisition, *SlDUR3* and *SlAMT1.1* for urea and ammonium uptake, respec-

tively, and the long-distance transport of nitrogen from the roots by *SlNRT2.3* [18,27,31]. Nevertheless, there is very little information about other transporter genes involved in the distribution of nitrogen species in the plant, especially when nitrogen availability is limited. We identified a group of new N transporters in tomato that showed increased expression under suboptimal conditions (4 mM N). We found that nitrate transporters *SlNPF2.6* and *SlNPF2.11* increased their transcript levels both in leaves and roots under conditions of limited N. Moreover, *SlNPF1.17*, *SlNPF7.6*, and *SlNPF7.3* were upregulated mainly in leaves. Interestingly, we also observed a significant increase in the urea transporter *SlDUR3* in both the roots and leaves. It has been reported that nitrate and urea significantly contribute to the remobilization of nitrogen from senescing leaves to sinks [33]. The differential responses between organs and N levels suggest that these transporters might display specific roles in the economy of nitrogen in the plant under N deficiency. Further work is required to unravel the molecular functions of the selected genes in the remobilization and distribution of nitrogen under nutritional stress conditions, and thus, as putative interesting transporters for tomato breeding under N deficits [79].

### 4.3. The Transcriptomic Analysis Revealed a Group of Candidate Transcription Factors Involved in Responses to N Deficiency

In the model plant Arabidopsis, different TFs have been identified that perform different functions in the control of nitrogen responses such as NIN-like protein 7 (NLP7) and NLP6 [109,110], NF-YA5 [111], TGA1/4 [112], bZIP1 [113], TCP20 [114], CDF3 [71], and HNI9/RTF1 [115]. Long-term metabolic and morphologic changes in Arabidopsis have been related to be regulated by TFs of the NF-YA and MYB families under starvation conditions [35]. Most of these transcription factors participate in a complex network involving further TFs in order to coordinate the transcriptional regulation of enzymes involved in the nitrogen metabolism [39,43]. However, very little information about transcription factors involved in nitrogen responses in tomato is available. In this study, we found several transcription factors that might be involved in the transcriptional response to suboptimal N supply in tomato. Of these, we have identified the homologues of Arabidopsis *TGA4* (Solyc04g054320), *NLP9* (Solyc08g013900), *HAT22* (Solyc02g063520), *NF-YA5* (Solyc08g062210), *NF-YA9* (Solyc01g008490), and *ARF18* (Solyc05g056040) genes, which are upregulated under conditions of limited N supply as reported in Arabidopsis (Table S5). Arabidopsis ARF18 is an auxin response factor that regulates the expression of nitrogen-related genes such as *NRT2.4*, *DUR3*, and *AMT1.2* [39]. Moreover, TGA4 is a key bZIP1 TF regulated by nitrate and involved in root growth [116] and HAT22 acts as a key regulator of the *ASN2* gene among others [117]. Notably, the nuclear factors NF-YA 5 and NF-YA 9 play an essential role in plants under starvation conditions [118]. Finally, it has also been reported that *NLP9* is upregulated under low N conditions in Arabidopsis [45] and GATA17, a member of the later N responders in the signaling network [43]. It is noteworthy that *NLP9* and *NF-YA5* are also differentially expressed in roots under N deficit conditions in tomato (Table S5).

Altogether, the expression data suggest differentiated networks operating in roots and shoots, thus indicating specific organ adaptations to N limitation. An ongoing functional characterization of the identified TF genes might shed some light on the regulation of tomato responses to a suboptimal N supply.

### 4.4. Impact of Suboptimal N Supply on Yield and Fruit Quality

Nitrogen deficiency decreases the tomato yield by lowering the production of the number of fruits and the fruit size as well as by negatively affecting the quality and taste [119]. In the case of tomato, fruit size is mainly determined by assimilate and water supply [120]. Accordingly, in our study, the lower photoassimilates available in the leaves such as sucrose and transported amino acids lead to smaller yield under limited nitrogen supply. Some studies into N withdrawal treatments indicate that source activity appeared to be more affected than sink activity and, therefore, vegetative growth was more reduced than reproductive [15,121]. Nevertheless, we observed a similar drop in both vegetative

and reproductive biomass as previously reported in tomato [71]. Furthermore, our analysis of NAE showed that the limitation in N supply led to a decrease, besides the N uptake efficiency by the plant, of the partition of N to the fruit, as yield N efficiency ($E_{N,Y}$) and N fruit contents ($C_{N,Y}$) were significantly affected.

Notably, the total elemental C concentration in the fruit was not altered by the limitation in nitrogen supply. Moreover, no changes were found in the soluble sugar contents and there were only minor changes in organic acids related to organoleptic fruit quality. Benard et al. [15] observed that a reduced N availability, in levels not causing N starvation, led to an increase in the content of fruit hexoses, a small decrease in citric acid but no changes in malic acid content, however, these changes were limited and dependent on the truss being considered. Furthermore, the changes in soluble sugars and organic acids brought about by nitrogen are usually triggered under conditions of excess of nitrogen [122]. Nevertheless, under our conditions, nitrogen limitation led to a drop in the total N content in fruits. It is known that increased N levels have a profound impact on the increased accumulation of glutamic and aspartic acids in the fruit [123]. Nevertheless, little is known about changes in specific amino acid contents under low nitrogen conditions. In this study, we found that the glutamic acid and GABA contents were significantly reduced in mature fruits under conditions of suboptimal N supply (Figure 8). Altogether, our data confirmed that the lower C and N photoassimilate availability impacts the yield by reducing fruit growth, although major carbon compound contents were not altered. In contrast, a significant drop in nitrogen compounds related to taste characteristics were produced. Furthermore, we determined phenolic compounds and flavonoids, major functional compounds in tomato fruits, since the transcriptomic data showed an over-representation of GO terms related to the phenylpropanoid biosynthetic pathways in leaves. However, no significant changes in the main hydroxycinnamic acids were observed in fruits grown under 4 mM N conditions, but there was increase in rutin content. Similar results were previously reported by Benard et al. [15]. It has been suggested that the tomato fruit is less reactive than leaves to the nitrogen supply, showing minor impact in the accumulation of flavonoids in the fruit [124]. Together, under suboptimal N supply conditions, we reported minor changes in key components of quality fruit components such as the main flavonoids as well as sugars and organic acids, although amino acid content decreased. Therefore, our results identified new target metabolic characters as markers in tomato breeding for higher NUE and as a means of preserving fruit taste under suboptimal N supply conditions.

## 5. Conclusions

To summarize, we described differentiated metabolomic and transcriptomic adaptation responses in roots and leaves in response to suboptimal N conditions. Although total amino acid contents decreased in leaves, glutamine and asparagine levels were maintained for partition to sustain growth, while surplus assimilated C was stored in roots. Accordingly, the comparative analysis of major co-expression clusters indicated organ-specific responses to the limitation in N supply. The transcriptomic analysis suggests that suboptimal N conditions promote changes in gene expression of many genes associated with C and N metabolism and remobilization that impact on plant metabolic rearrangement. In addition, our data suggest that both the mitochondrial alternative respiration and chloroplastic cyclic electron flux might be operating in long-term responses to nitrogen in tomato. Furthermore, we identified several transcription factors such as TGA4, NLP9, and NF-YA5, which might have a role in the regulation of the responses to suboptimal N supply.

Our results shed light on the transcriptomic and metabolomic responses of tomato at the organ level to a limitation of N fertilization compatible with sustainable conditions. The reduction in assimilated C and N available under limited N supply led to a reduction in the long-term of the vegetative and fruit biomass. However, the suboptimal N conditions impacted differentially on the C and N photoassimilates' partition to the fruits, and thus on their organoleptic quality. Minor changes were observed in sugars and organic acid contents in the fruits, whereas N compound amounts significantly dropped. The identified

processes and metabolic changes might be used for the targeted development of new varieties with improved responses to suboptimal N supply in breeding programs.

**Supplementary Materials:** The following are available online at https://www.mdpi.com/article/10.3390/agronomy11071320/s1, Figure S1: Effects of nitrogen supply levels on relative growth rate (RGR) in tomato, Figure S2: Effect of nitrogen supply levels on the uptake of micronutrients, Figure S3: Effect of nitrogen supply levels on amino acid contents in the leaves and roots of plants grown using 8 mM N and 4 mM N for 20 days, Figure S4: Verification of the transcriptomic analysis by RT-qPCR of selected DEGs from the C and N metabolisms, Figure S5: Gene co-expression clusters of roots and leaves under nitrogen limitation conditions (4 mM N), Figure S6: Metabolic gene expression changes under limiting N supply (4 mM N) analyzed by the MapMan tool, Figure S7: Effect of nitrogen supply levels on flavonoid and phenylpropanoid contents in mature red fruits, Table S1: Primers used in RT-qPCR analyses, Table S2: List of up- and downregulated genes in tomato leaves using 4 mM N when compared to 8 mM N, Table S3: List of up- and downregulated genes in tomato roots using 4 mM N when compared to 8 mM N, Table S4: List of upregulated and downregulated genes shared by leaves and roots, Table S5: List of DEGs corresponding to transcription factors in tomato leaves and roots using 4 mM N, Table S6: List of genes in root clusters under N limitation, Table S7: List of genes in leaf clusters under N limitation.

**Author Contributions:** Conceptualization, R.-V.M., J.M., and S.G.N.; Methodology, B.R.-M., E.G.M., J.C.-C., L.C., R.M., E.J.-B., L.Y., J.C. (Joaquín Cañizares), J.C. (Javier Canales), and S.G.N.; Analysis, R.-V.M., E.G.M., J.C.-C., V.G.-C., L.C., J.C. (Joaquín Cañizares), J.C. (Javier Canales), J.M., and S.G.N.; Investigation, B.R.-M., R.-V.M., E.G.M., J.C. (Javier Canales), J.M., and S.G.N.; Writing—original draft preparation, B.R.-M., R.-V.M., J.M., and S.G.N.; Writing—review and editing, R.-V.M., L.C., J.C.-C., E.G.M., J.M., J.C. (Javier Canales), and S.G.N.; Funding acquisition, R.-V.M., J.M., and S.G.N. All authors have read and agreed to the published version of the manuscript.

**Funding:** This study was supported by grants from The National Institute for Agriculture and Food Research and Technology (CSIC-INIA) (RTA2015-00014-c02-00 to JMA and RTA2015-00014-c02-01 to SGN) and the Community of Madrid (AGRISOST-CM S2018/BAA-4330 to JMA). We also want to acknowledge the "Severo Ochoa Program for Centers of Excellence in R&D" from the Agencia Estatal de Investigación of Spain (Grant SEV-2016-0672) for supporting the scientific services used in this study. J. Canales was supported by the Agencia Nacional de Investigación y Desarrollo de Chile (ANID, FONDECYT 1190812) and ANID-Millennium Science Initiative Program (ICN17-022).

**Data Availability Statement:** The data presented in this study are available on request from the corresponding author.

**Conflicts of Interest:** The authors declare no conflict of interest. The funders had no role in the design of the study; in the collection, analysis or interpretation of data; in the writing of the manuscript, or in the decision to publish the results.

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
