# Peer review of "Integrative Transcriptomic and Metabolomic Analysis at Organ Scale Reveals Gene Modules Involved in the Responses to Suboptimal Nitrogen Supply in Tomato"

_agronomy, doi:10.3390/agronomy11071320_

Round 1
Reviewer 1 Report
Very interesting study performed at laboratory scale and with different scales from trails to pots. It brings a first limitation of the study: the link between results of both experiments. Indeed, it might be challenging to combine the results and to draw conclusions based on studies with different scales. It should be referred in the manuscript.
In the pot experiment, it sounds like you use a destructive sampling. is it correct? if so, explained it in the M&M section. Also give more details about the trial management (weeds, irrigation...)
I will appreciate to have some few technical conclusions that should be use as "bring home" message for farmers, even if results were obtained at laboratory scale
Author Response
Dear reviewer,
We appreciate the changes and suggestions proposed to improve the manuscript. Here we addressed all the points that you have raised in your review of the running document. The manuscript has also been revised by an English native speaker.
We hope you will find that the new version meets the requested standards.
Best regards,
SG Nebauer
Reviewer Comments:
Very interesting study performed at laboratory scale and with different scales from trails to pots. It brings a first limitation of the study: the link between results of both experiments. Indeed, it might be challenging to combine the results and to draw conclusions based on studies with different scales. It should be referred in the manuscript.
Thank you very much for your suggestions and comments. We reported a decrease in N and C assimilation under suboptimal N conditions during the vegetative stage of growth that led to a reduction in the plant growth. Furthermore, organ specific changes in the C and N metabolites contents were triggered by the limitation in N supply. When plants were long-term cultivated under the same N limited fertilization conditions, the drop in photoassimilates availability also resulted in reduced yield when compared to control conditions. In addition, it was observed that the changes in partition of assimilates impacted differently on C and N compounds. Fruits maintained the contents of main compounds related to organoleptic quality, such as sugars and organic acids contents, whereas N compounds amounts, such as glutamic acid and GABA, significantly dropped. These different experiments have been addressed in the text (e.g. “We reported major transcriptomic and metabolomic changes in response to the limitation in N supply which led to a significant reduction in plant growth rates. We further wanted to assess the effect of suboptimal N conditions on the use of C/N compounds in the plant during the reproductive growth stage and the impact on the production of fruits.” (Section 3.7), and highlighted in the conclusions section.
Please take a look in the manuscript.
In the pot experiment, it sounds like you use a destructive sampling. is it correct? if so, explained it in the M&M section. Also give more details about the trial management (weeds, irrigation...)
We started the experiment growing the plantlets in trays filled with vermiculite to facilitate root sampling and transplanting. After 15 days, most plantlets were transferred from the trays to 1 L pots for the experiment in the growth chamber under sufficient and limiting N supply conditions (8 and 4 mM N, respectively) for 45 days. A subset of plants were transferred to 15 L pots and grown in the greenhouse for yield determinations under both N supply levels. Destructive sampling was performed for the 45 days experiment in growth chamber, and 50 to 20 plantlets were taken at each date. Standard tomato management and maintenance practices in the greenhouse were undertaken.
We changed the materials and methods section to clarify the procedures. Please take a look at the new version of the section.
I will appreciate to have some few technical conclusions that should be use as "bring home" message for farmers, even if results were obtained at laboratory scale.
Thanks for the point. This is a great suggestion. We have included a paragraph at the end of the Conclusions section addressing your suggestion.
Reviewer 2 Report
The manuscript includes some interesting results, however need more focus, and additional data analyses. teh manuscript completely lacks the integrated analyses of the metabolomics and transcriptome results.
The Materials and methods section need editing. It only gets clear at the later sections how many replicates were used in the various analysis steps, how are the analyses steps related to each other and how analyses of tissues and samples are related in the subsequent analyses.
Results section is imbalanced with more detailed results coming from the phenotypic analyses while the RNAseq results are not analysed properly and the various analysis sections are not integrated. It is not clear if the DEGs analyses and the subsequent GO overrepresentation analysis were performed using the separate lists of up-and downregulated genes for each tissue separately. It also needs clarification if there was any filtering performed and how signficiant analyses was performed. The results section also includes paragraphs, sentences that should be transferred to the discussion.
The different analysis parts are not evaluated properly. The data set has more value than mined and interpreted in the manuscript. Analysis methods such as gene set enrichment analysis of the up- and down-regulated gene lists, and multivariate analysis methods to identify gene module-trait relationships, define key genes with most signficant contribution to the traits, and especially integrated analysis methods to understand relationships between the observed metabolomics and transcriptomics changes would increase the value of the study.
The authors also include a tomato - Arabidopsis comparison that does not add value to the paper, the data comparison is not useful and not adequate as they were collected from samples grown under different treatments and in different conditions.
Author Response
Dear reviewer,
We appreciate the changes and suggestions proposed to improve the manuscript. Here we addressed all the points that you have raised in your review of the running document. The manuscript has also been revised by an English native speaker.
We hope you will find that the new version meets the requested standards.
Best regards,
SG Nebauer
Reviewer Comments:
The manuscript includes some interesting results, however need more focus, and additional data analyses. teh manuscript completely lacks the integrated analyses of the metabolomics and transcriptome results.
Thank you very much for the comments. We agree with your point, since we analyzed separately both datasets. We have performed additional analyses to address the integration of the metabolomic and transcriptomic data (see below). Please take a look at the Results and Discussion sections.
The Materials and methods section need editing. It only gets clear at the later sections how many replicates were used in the various analysis steps, how are the analyses steps related to each other and how analyses of tissues and samples are related in the subsequent analyses.
Thank you for the suggestion. We’ve revised and changed the Materials and methods section following your comments. More information about the experiments, sampling data and related analyses has been included in the section. Please take a look at the Materials and methods section.
Results section is imbalanced with more detailed results coming from the phenotypic analyses while the RNAseq results are not analysed properly and the various analysis sections are not integrated. It is not clear if the DEGs analyses and the subsequent GO overrepresentation analysis were performed using the separate lists of up-and downregulated genes for each tissue separately. It also needs clarification if there was any filtering performed and how signficiant analyses was performed.
Thanks for your comment. The physiological characterization of the effects of suboptimal N supply covered different aspects of plant growth, such as photosynthesis, nutrition, metabolism and biomass partition, and transcription analyses by RNAseq were performed in roots and shoots to identify genes and pathways involved. We agree with the comments, and additional technical details about the analyses has been included in Materials methods and Results sections, as well as additional analyses to a better integration with the metabolomics data.
Regarding the GO analyses, DEGs and GO analyses were performed using the separate list of up- and down- regulated genes for each organ (leaves and roots). Accordingly, we have modified the Figure 4 to make it more clear to the reader. Filtering and significance levels used for DEGs selection has been included in the text. Please take a look at the Materials and methods.
The results section also includes paragraphs, sentences that should be transferred to the discussion.
Thanks for the point. We have eliminated several sentences that either addressed a discussion of the results or compared them with previous studies. Please take a look at them in the new version of the manuscript.
The different analysis parts are not evaluated properly. The data set has more value than mined and interpreted in the manuscript. Analysis methods such as gene set enrichment analysis of the up- and down-regulated gene lists, and multivariate analysis methods to identify gene module-trait relationships, define key genes with most signficant contribution to the traits, and especially integrated analysis methods to understand relationships between the observed metabolomics and transcriptomics changes would increase the value of the study.
Thanks for the suggestion. We conducted additional analyses to complete the study. We performed co-expression network analysis to get insights about the patterns of gene expression, metabolites and processes affected by N limitation. Correlations among genes and group of genes with changes in primary metabolism compounds levels were also obtained to integrate transcriptomic and metabolomic datasets. Please take a look at section 3.6: Transcriptomic and metabolomic dynamics in roots and leaves under N limitation conditions in the Results section.
The authors also include a tomato - Arabidopsis comparison that does not add value to the paper, the data comparison is not useful and not adequate as they were collected from samples grown under different treatments and in different conditions.
Thanks for your coment. We were aware that the conditions of the Arabidopsis and tomato experiments are not exactly the same. Nevertheless, we performed the data comparison analysis since both data sets were obtained in samples of plants grown under long term suboptimal N conditions, and most studies on responses to N have been conducted under starvation conditions. We aim to identify conserved processes shared among species at suboptimal N levels at a long-term scale. Notably we identified some specific GOs in tomato, such as alternative oxidase and cyclic electron transport in photosynthesis. We highlighted them as newly described to be related to N responses. In any case, following your request, to simplify the text, we removed this analysis from the manuscript.
Round 2
Reviewer 2 Report
The manuscript has significantly improved after the suggested changes. The performed analyses are clear and supported by the figures and tables. The new sections included in the results and the discussion sections help the understanding and provide new insight in the identified genetic mechanisms affected by nitrogen stress.
There are some minor suggestions:
Materials and methods and results section 3.4: the mentioned fold change values for DEGs don't match, please, check them.
In 3.4 check the number of up and down regulated DEGs both for leaf and root tissues, the sum doesn't match with the total DEGs number, also check if the numbers in the Venn diagram are correct.
There are some minor typos in the document, please, check them.
Author Response
The manuscript has significantly improved after the suggested changes. The performed analyses are clear and supported by the figures and tables. The new sections included in the results and the discussion sections help the understanding and provide new insight in the identified genetic mechanisms affected by nitrogen stress.
Dear reviewer,
Thank you very much for the comments, and your help to improve the manuscript.
There are some minor suggestions:
Thank you for the suggestions. We’ve revised the text and changed it following your comments:
Materials and methods and results section 3.4: the mentioned fold change values for DEGs don't match, please, check them.
We have corrected the fold change values in the results section.
In 3.4 check the number of up and down regulated DEGs both for leaf and root tissues, the sum doesn't match with the total DEGs number, also check if the numbers in the Venn diagram are correct.
We have changed the total number of DEGs in the text, since the up-regulated and down-regulated figures were correct in the text and Figure 4.
There are some minor typos in the document, please, check them.
We’ve revised the manuscript and typos have been removed (e.g. gene names in italics, subscript in “log2FC”,…)
We hope you will find that the new version meets the requested standards.
Best regards,
SG Nebauer